# Post Treatment Sexual Function and Quality of Life of Patients Affected by Cervical Cancer: A Systematic Review

**DOI:** 10.3390/medicina59040704

**Published:** 2023-04-04

**Authors:** Stefano Cianci, Mattia Tarascio, Martina Arcieri, Marco La Verde, Canio Martinelli, Vito Andrea Capozzi, Vittorio Palmara, Ferdinando Gulino, Salvatore Gueli Alletti, Giuseppe Caruso, Stefano Restaino, Giuseppe Vizzielli, Carmine Conte, Marco Palumbo, Alfredo Ercoli

**Affiliations:** 1Obstetrics and Gynecology Unit, Department of Human Pathology of Adult and Childhood “G. Barresi”, University of Messina, 98124 Messina, Italy; 2Obstetrics and Gynecology Unit, Department of Woman and Child, Azienda Sanitaria Provinciale di Catania, 95124 Catania, Italy; 3Department of Biomedical, Dental, Morphological and Functional Imaging Science, University of Messina, 98122 Messina, Italy; 4Department of Medical Area (DAME), Clinic of Obstretics and Gynecology “Santa Maria della Misericordia”, University Hospital Azienda Sanitaria Universitaria Friuli Centrale, University of Udine, 33100 Udine, Italy; 5Obstetrics and Gynecology Unit, Department of Woman, Child and General and Specialized Surgery, University of Campania Luigi Vanvitelli, 80138 Napoli, Italy; 6Department of Medicine and Surgery, University Hospital of Parma, 43125 Parma, Italy; 7Department of Obstetrics and Gynaecology, Azienda di Rilievo Nazionale e di Alta Specializzazione (ARNAS) Garibaldi Nesima, 95124 Catania, Italy; 8Obstetrics and Gynecology Unit, Department of Woman and Child, Ospedale Buccheri La Ferla Fatebenefratelli, 90123 Palermo, Italy; 9Department of General Surgery and Medical Surgical Specialties, Gynecological Clinic, University of Catania, 95123 Catania, Italy

**Keywords:** cervical cancer, sexual function, quality of life, sexual dysfunction

## Abstract

*Introduction*: The aim of this study is to analyze the available scientific evidence regarding the quality of life (QoL) and sexual function (SF) in patients affected by cervical cancer (CC) after surgical and adjuvant treatments. *Materials and Methods*: Preliminary research was conducted via electronic database (MEDLINE, PubMed and Cochrane Library) with the use of a combination of the following keywords: SF, QoL, and CC. The principal findings considered in the present review were the study design, the number of patients included in each study, the information about the malignancy (histology and stage of disease), the questionnaires administered, and the principal findings concerning SF and QoL. *Results*: All studies were published between 2003–2022. The studies selected consisted of one randomized control study, seven observational studies (three prospective series), and nine case control studies. The scores used were focused on SF, QOL, fatigue, and psychological aspects. All studies reported a decreased SF and QOL. The most developed questionnaires were the European Organization for Research and Treatment of Cancer Quality of Life Questionnaire (EORTC QLQ-C30), the Female Sexual Function Index (FSFI), the Hospital Anxiety and Depression scale (HADS), and the Female Sexual Distress Scale (FSDS). *Discussion*: All studies reported a decreased SF and QOL. In addition to the perception of body image, several factors coexist in influencing the outcomes such as the physical, hormonal, psychological. *Conclusions*: Sexual dysfunction after CC treatment has a multifactorial aetiology which negatively affects the quality of life. For these reasons, it is important to follow and support patients with a multidisciplinary team (doctors, nurses, psychologists, dieticians) before and after therapy. This type of tailored therapeutic approach should become a standard. Women should be informed about possible vaginal changes and menopausal symptoms after surgery and on the positive effects of psychological therapy.

## 1. Introduction

Cervical cancer (CC) is the fourth most common neoplasm among women, with a highest incidence in young women [1]. In 2019, a total of 565,541 new cases and 280,479 deaths caused by cervical cancer were registered [2]. Persistent human papillomavirus infection is the most important cause leading to cervical cancer. The incidence and mortality of CC differs geographically and is related to the Human Development Index (HDI). Countries with low and middle HDI have worse rates than those with high HDI [3].

In recent years, thanks to the strengthening of screening programmers, through the Pap Test, the Human Papilloma Virus (HPV) DNA test and the HPV vaccination, there has been a significant decrease in incidence and an increase in survival rates for CC: [4,5,6,7,8,9]. 

In May 2018, the World Health Organization (WHO) has launched a global call for action to eradicate cervical cancer [10,11,12,13]. The WHO recommended two doses of the HPV vaccine for girls aged 9 to 14. Vaccination is primarily recommended for younger individuals, preferably before the beginning of sexualactivity. The HPV vaccine’s objective is to prevent HPV infection and HPV related-cancer [14]. The WHO’s strategy for the elimination of cervical cancer has a target of 90% of all adolescent girls being vaccinated against HPV by 2030. Nevertheless, despite the efforts to strengthen screening and vaccination programs, the goal is still far from being achieved [15,16,17,18,19,20,21,22,23]. Currently, radical hysterectomy (RH), radiotherapy, chemotherapy, or a combination of both are considered as the standard treatment, especially for early stage CC [24]. However, medical and surgical procedures could be related to severe post-treatment dysfunctions with regard to sexual, intestinal, and vesical functions [25,26,27]. 

A significant percentage of CC patients experienced frequent sequelae after treatment, such as decreased sexual desire, vaginal dryness, dyspareunia, depression, and anxiety. These aspects acquire greater relevance considering that the most CC patients are relatively young. Currently, the quality of life (QoL) of cancer patients has gained more relevance than in the past, and the efficacy of treatment is not the only aspect to evaluate. In this context, especially for young patients, achieving regular sexual function is extremely important [28,29]. However, the limits with CC are that the treatment options to reduce sexual dysfunction and menopausal symptoms (i.e., local or systemic estrogen, nonhormonal treatments, laser therapy, lubricants) are extremely limited and are not always effective for this subgroup of patients [30,31,32,33,34].

Many recent reviews focused on quality of life and the sexual function of patients with cervical cancer and have only evaluated the sexual dysfunctions of patients undergoing chemo- radiotherapy plus interventional radiotherapy [35,36,37]. The studies currently available in the literature are poor and heterogeneous, and their results are often conflicting; therefore, no definitive recommendations can be formulated. 

Furthermore, it is unclear whether the dysfunction is attributable to surgical sequelae, radiotherapy, chemotherapy, or psychological aspects related to cancer. In fact, psychological factors alone caused by the cancer diagnosis may also negatively influence sexual functioning independently of the type of treatment or stage of cancer.

During oncological evaluation and follow-up, information and communication about sexual issues is often lacking. The aim of this study is to analyze the available scientific evidence regarding QoL and especially the sexual function (SF) of patients affected by CC after surgical and/or medical and radiological treatment.

## 2. Materials and Methods

This systematic research has been performed in agreement with the Preferred reporting items for Systematic reviews and Meta-analysis statement (PriSMa). A comprehensive literature research on electronic databases (MEDLINE, PubMed and Cochrane Library) was conducted from inception until July 2022. 

The primary research strategy was identified with the use of a combination of the following medical relevant headings terms (MeSH) and keywords: “sexuality and cervical cancer”, “sexual function and cervical cancer”, “cervical cancer and sexual dysfunction”, “sexuality and gynecological cancer”, “sexual dysfunction and gynecological cancer”, “sexual dysfunction and cervical cancer”, “cervical cancer and Quality of Life”, and “gynecologic cancer and Quality of Life”. Studies evaluated that were not in line with the aim of the study, case reports, papers based on animal models or laboratory studies, and non-English language articles were excluded. 

The electronic research and the eligibility of the studies were independently assessed by two of the authors (MA, MT). In addition, references in the included articles were reviewed to identify additional eligible articles. Differences were discussed with a third reviewer (SC) for data extraction.

The main findings considered in the present review were the study design, the number of patients included in each study, information about the pathology (histology and stage of disease), the questionnaires administered, and the principal findings concerning SF and QoL. 

The data extracted were the author, year of publication, number of patients included, study population characteristics (age, comorbidity, social status), main outcomes, main findings, and sexual and quality of life questionnaires. After the initial research, a total of 12177 titles were extracted from the Pubmed database, MEDLINE, and Cochrane Library using the keywords previously mentioned. After the first revision, 122 unique studies were extracted. After further revision, 25 articles were selected, out of which 18 studies were ultimately eligible for the present review following a full text evaluation. The selection process is illustrated in Figure 1. Considering the heterogeneity of the studies in terms of methodology, population, and the different score administered, a cumulative analysis was not applicable. However, the main outcomes of different studies were analyzed.

## 3. Results

### 3.1. Study Characteristics

A total of eighteen studies treating both SF and QoL in patients affected by CC were extracted. All studies were published between 2003 and 2022 and are reported with the main findings and characteristics shown in Table 1.

The selected studies were represented by one randomized study, eight observational studies (three prospective studies), and nine case control studies. Different validated scores were used to assess SF and QoL in the different studies; however, the most frequently administered questionnaires were the European Organization and Treatment of Cancer QLQ-C30 (EORTC QLQ-C30), the Quality of Life Questionnaire for Cervical Cancer (EORTC QLQ-CX24), the Female Sexual Function Index (FSFI), the Hospital Anxiety and Depression scale (HAD S), the Sexual Function-Vaginal changes Questionnaire (SVQ), and the Female Sexual Distress Scale (FSDS).

### 3.2. Study Descriptions

The study of Kirchheiner et al. [38] reported the largest series, with 1045 cases, and considered women affected by CC from stage FIGO I to IVB who underwent to chemoradiotherapy and image-guided adaptive brachytherapy. The aim of the study was to prospectively investigate sexual outcomes before and after treatment. The primary objective was to report patient sexual activity, vaginal functioning problems during intercourse, and sexual enjoyment within the first 5 years of follow-up. They assessed the SF at baseline and every 3 months during the first year, and thereafter every 6 months in the second and third year, and yearly after that using EORTC-QLQ-CX24. A total of 622/1045 (60%) patients were sexually active after treatment, but many of them complained of different aspects of SF during follow up: 39% reported vaginal dryness, 38% reported vaginal shortening, 35% reported vaginal tightening, and 34% reported pain during sexual intercourse. 

These symptoms were negatively correlated with sexual enjoyment (*p* ≤ 0.001). Regular hormonal replacement therapy (HRT) was associated with a significant improvement in vaginal dryness (*p* = 0.015), vaginal shortening (*p =* 0.024), and pain during intercourse (*p* = 0.003).

Novackova et al. [39] compared QoL and SF (EORTC QLQ-CX24, EORTC QLQ-CX30, FSFI) in patients with early-stage CC before and after a nerve sparing radical hysterectomy. The results showed a statistically significant decrease in arousal, orgasm, desire, lubrication, satisfaction, and pain after surgery. The results of the EORTC QLQ indicated a decrease in sexual functioning after surgery, but no changes in sexual activity, sexual enjoyment, or concern. 

The study by Park et al. [40] reported 860 cases (women with a history of CC, stage FIGO I to IVA) and 494 controls. The authors investigated the QoL and SF through EORTC QLQ-C30. They reported that women affected by CC had greater impaired social functioning and more severe urinary symptoms, constipation, and diarrhea than controls (*p* < 0.01). Moreover, CC survivors had more severe menopausal symptoms and worse self-body image. Furthermore, SF was impaired, as women with CC had more anxiety about sexual performance and worse vaginal functioning (*p* < 0.01). The sub-analysis performed, based on the treatment received, demonstrated that sexual dysfunction was more severe in women who underwent radiotherapy. Moreover, patients who received both surgery and radiotherapy experienced a higher decrease in SF than women that only had surgery. In addition, chemotherapy was linked to dyspareunia (odds ratio [OR]), 1.6; 95% CI, 1.4–2.0), anxiety about sexual performance (OR, 1.7; 95% CI, 1.4–2.1), and insufficient vaginal lubrication (OR, 1.3; 95% CI, 1.1–1.6) in a multivariate analysis. Finally, a high association between sexual problems and a QOL decrease in CC cancer survivors was recorded. 

Jensen et al. [41] prospectively compared 173 patients with early-stage CC, negative lymph node submitted to radical hysterectomy (RH), and pelvic lymphadenectomy to 328 controls. The authors assessed SF using the SVQ, a validated self-assessment questionnaire developed to evaluate sexual and vaginal disorders after gynecologic cancer. CC patients experienced low or no sexual interest and severe lack of vaginal lubrification more frequently than controls during the first 2 years after surgery. 

Compared with controls, CC patients experienced significantly severe orgasmic dysfunction and uncomfortable sexual intercourse due to a reduced vaginal length during the first 6 months after radical hysterectomy, reporting severe dyspareunia during the first 3 months (RR = 3.5, CI = 1.2–10.2 [5 weeks]; RR = 2.5, CI = 1.0–6.5 [3 months] and lower sexual satisfaction during the 5 weeks after surgery. However, six months after surgery, the proportion of patients who were sexually active was similar to that of the control group of the corresponding age. Comparing SF retrospectively before and after surgery, cancer survivors complained of a higher level of sexual dysfunction 12 months after surgery; 91% of CC patients were sexually active before and after surgery, with a decrease in sexual activity frequency (*p* = 0.008) and sexual interest.

Ljuca et al. [42] evaluated 35 patients with advanced CC who had exclusive chemo-radiotherapy to assess their SF before and after treatment using the EORTC-QLQ-Cx, a questionnaire specifically elaborated for CC. Women answered questions about SF in the period before treatment and 12 months after the end of chemo-radiotherapy. After therapy, vaginal dysfunction was statistically reduced (*p* < 0.0001). However, there was no significant difference in SF. Fourteen patients (40%) did not have sexual intercourse either before or after irradiation. Dyspareunia was significantly reduced (*p* = 0.009) after therapy, while vaginal function in terms of lubrification improved significantly.

Bakker et al. [43] analyzed 194 sexually active patients and 58 sexually inactive CC survivors, who underwenteither RH with pelvic lymphadenectomy or radiotherapy. Sexual distress was reported by 38% of all participants associated with sexual pain, anxiety, and body image concerns. Sexually inactive cervical cancer patients were significantly older, in a stable and long-term relationship, were relatively more often diagnosed with FIGO stage IIB or higher, and were more often treated with RT compared to sexually active women.

The severity of vaginal sexual symptoms reported by sexually active participants was significantly different according to the treatment received: patients who underwent radiotherapy/brachytherapy reported significantly higher vaginal discomfort than women who underwent RH (*p* = 0.004) and RH/radiotherapy (*p* = 0.008). However, no differences were recorded compared to women treated with RH/radiotherapy/brachytherapy (*p* = n.s.). Higher levels of sexual distress were significantly correlated to vaginal symptoms such as dryness, sexual pain, worry, anxiety, depression, and body image concerns.

Aerts et al. [44] prospectively compared the SF of 31 women affected by CC who underwent RH, 93 who underwent simple hysterectomy due to a benign gynecological condition, and 93 healthy controls. In the cases group, only 25 women completed the surveys at 6 months after surgery (25/50, 50%), 14 women completed it at 1 year (14/50, 28%), and 12 women completed it at the 2 year follow-up (12/50, 24%). No difference in SF was recorded in CC survivors before and after surgery compared to women who underwent hysterectomy for a benign condition. However, the comparison with healthy controls, preoperatively and postoperatively, showed significantly more sexual dysfunction in the cancer group. Introital dyspareunia was significantly higher in women with CC before treatment (*p* < 0.01), and it persisted during the 2-year follow-up; furthermore, the intensity of orgasm and sexual arousal was significantly reduced at 1 year of follow up. CC survivors reported more deep dyspareunia (*p* < 0.01) and abdominal pain during intercourse at 2 years after treatment (*p* < 0.01) than healthy women. Moreover, worse psychological functioning before surgery and at 6 months after surgery were reported in the CC patients compared to healthy controls, as no differences were recorded at the 2-year follow-up. 

In CC patients, the quality of their before the surgery was better than in healthy controls, while it decreased during the first year of follow-up compared to the interval before surgery.

Correa et al. [45] investigated the SF in CC survivors (*n* = 33) in comparison to a control group (*n* = 37) of women without a history of cancer using the FSFI test. SF decreased among the CC survivors, and 59.5% of women reported that the diagnosis and/or treatments of CC interfered with their sexuality. No differences were noticed with regard to sexual frequency in active women among the two groups. The mean scores of the cancer group were statistically inferior (*p* < 0.05) than those of the control group in all of the FSFI domains and in the total score. The mean total score in cancer survivors was 21.72. A FSFI score lower than 26 is indicative of sexual dysfunction. The authors reported that 64.9% of CC women reported vaginal stenosis or shortening; 59.5% were sexually inactive, and among those sexually active, 80% complained of sexual dysfunction.

Lee et al. [46] compared the QoL and SF in sexually active CC survivors (*n* = 104), with 45 months of median interval from diagnosis, and the healthy women group (*n* = 104). The authors did not record any statistical difference in SF between the two groups. It should be noted that, during the enrollment, 38.2% (91/238) of CC survivors and only 10.6% (28/265) in the control group were excluded due to the absence of sexual activity within 3 months.

Fifty-two patients who underwent surgery for early CC cancer were enrolled by Carter et al. [47] to assess their SF and QoL. Among these patients, 33 patients underwent radical trachelectomy, while 19 underwent RH. Both groups preoperatively demonstrated scores suggestive of depression and distress; FSFI were below the mean cut-off, highlighting a sexual dysfunction; however, the mean score increased from 16.79 preoperatively to 23.78 at 12 months and 22.20 at 24 months, with no significant difference between the groups.

In a Swedish study by Hofsjo et al. [48], 34 patients treated for CC with radiotherapy, either primary or in combination with surgery and/or chemotherapy, and 37 healthy age-matched control women scheduled for benign gynecological surgery were included. All subjects completed a questionnaire designed to assess sexual function in CC patients.

The women with cancer complained of worsening SF. The highest relative risk (RR) was reported for insufficient vaginal lubrication (RR 12.6), vaginal elasticity reduction (RR 6.5), reduced genital swelling when sexually aroused (RR 5.9), and the reduction of vaginal length during intercourse (RR 3.9). The frequency of orgasm was also lower in CC patients. 

De Rosa et al. [49] investigated the role of ospemifene in CC survivors with vulvovaginal atrophy, focusing on their QoL and SF. Fifty-two patients were treated with ospemifene and were evaluated at baseline and after 6 months through the Vaginal Health Index (VHI), while the SF and QoL were measured by EORTC QLQ-CX24. After the treatment, each parameter of VHI significantly increased, primarily elasticity, fluid volume, epithelial integrity, moisture, and pH. In addition, SF was significantly improved, especially sexual activity, sexual vaginal functioning, body image, and sexual enjoyment. QoL partially changed at 6-month follow-up, and global health status and emotional and social functioning scores increased significantly, while general symptom scales did not change from the baseline data.

Plotti et al. [50] retrospectively evaluated QoL, urinary function, and SF in locally advanced CC patients previously who underwent neoadjuvant chemotherapy and RH type C2/type III and with at least 36 months of follow up. The authors administered the EORTC QLQ-CX24 Questionnaire, the EORT QLQ-C30, and the Incontinence Impact Questionnaire 7 to 90 patients. The global health status, referred to as the perception of well-being and QoL, was 71.72 (the score ranged from 0 to 100). CC survivors complained of gastrointestinal symptoms, mainly diarrhea (6% of patients) and constipation (75% of women). The results related to SF showed a good level of sexual enjoyment (75.6%), with a slight worsening of sexual activity. A total of 28% of women reported mild incontinence, although it was rarely disabling.

Bae et al. [51] reported a series of 137 CC patients. The authors assessed the SF by FSFI, depression by HAD S, and QoL by Functional Assessment of Cancer Therapy–General version 4 (FACT-G). They underlined that CC women reported sexual dysfunction (4.83 ± 4.16) and moderate to severe depression (11.08 ± 5.06) with a mean score of quality of life of 57.33 ± 8.47. The total score of SF ranges from 1.2 points to 36 points; higher scores imply higher SF. SF had a negative correlation with depression, while having a positive relationship with quality of life (*p* < 0.001) and physical well-being, social well-being, and functional well-being (*p* = 0.001). Instead, the correlation was not significative with psychological well-being (*p* = 0.223).

Serati et al. [52] compared SF in women who underwent RH with a control group of healthy women using the FSFI. They enrolled 38 patients treated for early CC and 35 controls. They also analyzed the possible impact of the surgical approach (n. 20 cases of laparoscopy vs. n. 18 cases of laparotomy) on SF. The FSFI score was significantly higher in the healthy controls compared to the cases. No significant differences were recorded between the laparoscopic and the laparotomic group (*p* = 0.30). In the laparoscopic group, 75% of women reported an impaired SF, 20% reported no change, and 5% reported an improvement post-RH; in the laparotomy group, the proportions were 55.5%, 33.4%, and 11.1%, respectively. In addition, at the time of FSFI administration, 14 (36.8%) women had not yet resumed sexual activity (8/20 (40%) in the laparoscopic group vs. 6/18 (33%) in the laparotomy group; *p* = 0.74). 

A unique published randomized control study [53] analyzed a nurse-led positive psychology intervention on SF, depression, and subjective well-being amongst patients submitted to RH for early-stage CC. A total of 91 patients were enrolled in three tertiary hospitals in Chongqing, China, and were randomly allocated to the intervention (*n* = 46) or control group (usual care, *n* = 45). The intervention group received a 4-week PERMA model-based multidisciplinary team (leading nurses, specialist nurses, gynecologists, psychological counsellors, and rehabilitation physiotherapists) psychological intervention in addition to the usual care. The PERMA model is based on five independent elements: positive emotion (P), engagement (E), relationships (R), meaning (M), and accomplishment (A).

The authors analyzed the FSFI, the Self-rating Depression Scale, and the index of well-being to assess their aims at baseline and at 3 and 6 months after intervention. The SF was significantly improved in the intervention group (mean difference: −3.95, *p* = 0.005 at 3 months post-intervention; mean difference: −4.36, *p* = 0.001 at 6 months post-intervention), compared to the control group. In addition, the levels of depression and well-being in the intervention group were better in the intervention group (*p* < 0.05).

Cerentini et al. [54] tested the use of vaginal dilators in cervical cancer patients who underwent brachytherapy. However, the use of these tools did not increase the dimension of the vagina (length: *p* = 0.111, width: *p* = 0.484) within the first 3 months after the end of radiotherapy treatment.

Stanca et al. [55] used the standardized questionnaires (EORTC) to assesses the outcomes of QoL and SF after radical hysterectomy in CC. The survey showed a decrease of sexual function, activity, enjoyment, and an increase in sexual concern, together with menopausal symptoms.

### 3.3. Main Findings

All studies showed a general decrease of SF in patients with CC. However, the reasons for the increase in sexual dysfunction were different. The stage of the disease and the consequent therapeutic treatment were the two most important factors for sexual dysfunction. However, most studies had a sample with different cancer stages and with patients who underwent different treatments.

Several studies, such as those by Park, Stanca, Plotti, Bakker and Serati et al. [40,43,50,52,55] reported a significant worsening in patients who received both surgery and adjuvant therapy compared to surgery alone. In these studies, there was no difference between the surgery approaches. This aspect was also confirmed in the study by Carter [47], where there were no differences in SF in the patients who underwent radical trachelectomy or radical hysterectomy. This aspect was not observed in the Correa study [45], and this was likely due to the reduced number of women who underwent surgery. Aerts and Hofsjö et al. [44,45,46,47,48] reported sexual disfunction in CC patients who underwent surgery, but no difference if comparing the cancer group with the benign gynecological disease group. Serati et al. [55] did not report any benefit in CC patients between the laparoscopy and laparotomy approaches. Novackova and Jensen [39,41] showed no changes in sexual activity, enjoyment, or worry in patients who underwent surgery without adjuvant therapy. Bae et al. [51] showed that sexual dysfunction was related to a high level of depression. The scores of depression were higher for patients who underwent both types of chemoradiotherapy, althoughthere was no difference on the disease stage. Only Ljuca et al. [42] showed an improvement of vaginal function after adjuvant therapy; however, the survey consisted of a small number of patients. Plotti and Shi [50,51,52,53] suggested that SF improved over time in both the surgery and chemo or radio-therapy groups.

## 4. Discussion

Sexual dysfunction after CC treatment has a multifactorial aetiology which negatively affects the QoL. The most frequent causes of sexual dysfunction are psychological factors such as depression, loss of interest in sex, anxiety, and the presence of surgical scars that affect body self-image. Another important psychological factor is the fear of cancer recurrence that affects all aspects of life [56,57,58]. Furthermore, physical disorders such as asthenia, vaginal dryness, dyspareunia, and vaginal shortness are likely to occur after CC treatment, contributing to sexual distress (Table 2).

QoL encompasses several aspects of a person’s life in addition to a healthy sexual life, such as work activities, friendship, hobbies, and sport. The balance between all factors determines a healthy equilibrium, and therefore a positive spectrum of life quality. Conversely, factors negatively influencing the right balance can determine a chain reaction that contributes to a decrease in wellness. A cancer diagnosis and consequent surgical and/or medical treatments could be one of the most relevant destabilizing factors. Considering the CC characteristics and its development in the genital area, the impact on SF becomes more relevant, especially in young women with an active sexual life.

As previously mentioned, patients affected by CC can experience relevant changes in different aspects of life, such as the psychological state (such as anxiety and stress), body appearance, malnutrition (particularly during adjuvant therapy), relationship misunderstandings with their partner, and functional and anatomical problems. Undoubtedly, the impact of CC on QoL, as reported in the literature, could be different depending not only on the treatment adopted, whether surgical, medical, and/or radiation therapy but even by age, personal characteristics, and social aspects.

The surgical approach that is most adopted for CC treatment is the RH. The procedure consists of the resection of the parametrium and part of the vagina. The radicality of the surgery depends on the disease stage. This surgical approachis often associated with post-surgical complications, due to the radical nature of the surgery and neurological dysfunction. The nerves’ damage may cause: bladder function, intestinal dysfunction, vaginal shortness and dryness. The CC surgery could often be associated with chemotherapy and radiotherapy, which could enhance the genital dysfunction. All of these physical aspects associated with the psychological status could significantly interfere with SF and QoL.

The surgical approach could be endoscopic (laparoscopic or robotic) or laparotomic, depending on the different clinical and oncological aspects and the hospital’s internal guidelines.

Following the Laparoscopic Approach to Cervical Cancer (LACC) trial [59], the surgical approach to CC has been changed, and the current preference, especially for tumour size > 2 cm, is the laparotomic approach. In fact, this randomized trial, which compared minimally invasive surgery with laparotomy for CC treatment, demonstrated that the overall survival and disease-free survival of patients affected by CC undergoing laparotomy surgery is more effective compared to the endoscopic approach, even if the laparotomic approach was likely to interfere significantly with QoL and SF [60]. This aspect was confirmed in a study by Gueli Alletti et al. [61], demonstrating that that the body self-image and, consequently, QoL, decreases more rapidly and severely in cancer patients treated with laparotomy compared with a group of a patients treated with minimally invasive surgery (*p* = 0.004) (*p* = 0.002).

Several studies in the literature have evaluated the advantages of endoscopy both for benign pathology and oncology [62,63,64].

Minimally invasive surgical techniques employed on gynaecologic cancers, such as ovarian and endometrial cancer, have demonstrated advantages in improving surgical and aesthetic outcomes in selected cases [65,66,67], with promising results. The endoscopic approach is actually considered to be the gold standard for early-stage endometrial cancer treatment, and has several applications, even for ovarian cancer; however, this surgical approach is not always feasible for all oncological cases, especially for CC treatment after the LACC trial.

The limitation in the choice of surgical approach, based on the literature data, could contribute to a further increase in sexual dysfunction. Moreover, given the young average age of CC diagnosis and the impact of invasive treatments, the SF and QoL aspects need to acquire more relevance, and they should be considered in all phases of the treatment path, from counselling to follow-up [68,69].

Radiation therapy has an impact equivalent to surgery on SF, reducing the vaginal length and enhancing the vaginal dryness due to an actinic reaction.

The CC patients frequently undergo both surgery and radiation therapy, with the consequent reduction of the length and elasticity of the vagina. Even for patients submitted to exclusive chemo-radiation, the effects of brachytherapy are often superimposable.

As a consequence, for all these patients, sexual intercourse may be difficult and painful, and most of them are not able to achieve fulfilling sexual intercourse.

Recently, Beltran et al. [70] investigated the quality of life, sexual function, and satisfaction through the EORTC, FSFI and GRISS questionnaires, and compared the results to healthy women. This control studies reported a series of 66 patients analysed. The results showed a decrease of sexual function and quality of life (*p* < 0.05) in all questionnaires. This control study confirmed the negative impact that cervical cancer with medical therapy can have on women. However, a limit of the study was that the main treatment consisted of a combination of chemotherapy, radiotherapy, and brachytherapy, and did not include patients who underwent surgery. In fact, it is currently unclear whether the sexual dysfunction is attributable to surgical sequelae, chemo-radiotherapy, both, or all of these therapies.

Cancer survivors experience a loss of interest in sex and a consequent decrease in sexual activity, especially immediately after the treatment. In fact, the negative impact on the woman’s life is higher at the beginning of the treatment plan [71]. The most reassuring aspect, as reported by Carter J. et al. [72], is the fact that the time elapsed from treatment is a positive factor, and most patients report an improvement in SF.

All of these influencing factors are relevant, and it is important to know them in order to be able to attend to them promptly and effectively.

For these reasons, as confirmed in the study by Galicaa J. et al. [73], is important to follow and support CC patients with a multidisciplinary team (doctors, nurses, psychologists, and dieticians) before and after therapy. Women should be informed about possible vaginal changes and menopausal symptoms after treatments, in addition to the positive effects of psychological support, clinical nutrition, and physical activity on their life including SF [74,75]. The CC patients should be also counselled about the safe use of local estrogen therapy and systemic hormonal replacement therapy, when feasible, to improve their SF.

This type of tailored therapeutic approach should become a standard treatment, as the principal goal is not only to fight the pathology but also to achieve a good QoL for patients.

This article aims to give an overview of the latest findings related to QoL and SF in CC treatment. The most important limitation of this review is related to the heterogeneity of the available studies. The available studies present different designs and a significant heterogeneity of patients who underwent different treatments. Moreover, the studies were conducted during different periods, from 2003 to 2022, and the treatments adopted are not sufficiently comparable due to the evolution of therapeutic strategies over time. Finally, the most significant shortcoming is the fact that most of the studies had a small number of patients, and the tests used are different and not always comparable. It is clear from all studies that CC has an impact on Qol and SF, but it is unclear whether it is more related to a psychological condition or to a consequence of medical or surgical treatments. Prospective studies using standardised questionnaires are needed to reach definitive conclusions.

The current study, however, despite the limitations previously reported, did result in some important findings. The impact of medical (chemoradiation therapy) vs. surgical treatment on SF remains controversial because the studies reported different conclusions (Table 1).

Some authors have suggested that there are no significant differences in SF and QoL depending on the surgical approach, and that most CC cancer patients report an improvement of SF and the disappearance of vaginal problems over time. However, based on the results of most studies, the combined approach of surgery and radiotherapy seems to have the most negative impact on SF [76].

A key aspect remains that, after CC diagnosis, both patients and their families and care professionals should focus not only on treating cancer, but must not forget that an adequate quality of life must be maintained.

In this field, the personalisation of cancer treatment is essential, and must take into account such factors as age, social status, and expectations. Furthermore, it is important to provide psychological and psychophysical support, not only during treatment but also afterwards. Moreover, several studies have proven the importance of different types of support, in addition to psychotherapy including: exercise, yoga, and biofeedback [77,78,79,80,81].

Over the years, thanks to technological innovations, surgical and medical treatment approaches have reduced their physical impact, while maintaining the same levels of efficiency. Although this is an important point, it cannot be considered sufficient to overcome the entire impact on the QoL and SF of cancer patients, which can only be achieved with constant support. It is relevant that sexual disfunction after CC treatment tends to improve over time [82,83]. Evidence from studies of other gynaecological malignancies has confirmed these findings, reporting that sexual function improves with increasing time since the cancer diagnosis [84,85]. The importance of the SF of cancer patients is still not sufficiently considered by the majority of care professionals, who forget to investigate and treat it appropriately during cancer treatment and follow-ups. Women with CC are often young, and the absence of a satisfactory sex life can affect their personal life and relationships, with severe consequences to their QoL.

Indeed, it is important to understand that the psychological aspects related to cancer treatment and the consequent impact on QoL and SF are related to several factors that might be different in patients, such as age, social status, work, geographical area, religion, etc. Consequently, it is crucial to adapt the treatment according to the patient’s characteristics. A relatively new factor that should be taken into account is cancer prevention and screening. Even though screening programmes and cancer prevention strategies are still relatively ineffective for most cancer diseases, CC has a great potential for reduction through prevention and early detection. This should publicised more widely, as it seems to be the key strategy in terms of fighting CC.

In 99% of cases, CC is caused by HPV infection [86,87,88], as demonstrated by studies focused on countries such as Australia and Rwanda. A screening program is the best strategy to decrease CC incidence in the short term, and HPV vaccination programs represent the best strategy to eradicate CC in the long term [89,90,91,92,93]. According to the World Health Organization (WHO), the primary goal must remain the eradication of CC through the strengthening of vaccination and screening programmes. However, due to a heterogeneous adherence to HPV vaccination and Pap-testing in different countries, it remains a difficult goal to achieve [94,95,96]. The sensible reduction of QoL and SF after CC treatments will continue to be an issue in the coming years. Therefore, new randomized clinical trials are needed in order to investigate which treatment option may be the most effective for the management of vaginal symptoms related to CC treatment. Several studies have shown the benefit that hormonal therapies such as CO2 laser therapy, hyaluronic acid, and vaginal dilators may have on the improvement of vaginal symptoms [97,98,99,100,101,102,103,104,105,106]. Although there are no trials with large series’ that compare all of these therapies in patients with CC, it is considered essential, in alignment with the WHO goal on the elimination of HPV-related CC, to also focus therapeutic research on the improvement of QoL and vaginal symptoms that would help all women after treatment. Therefore, further studies are needed to clarify the factors contributing to the decrease in SF and QoL in CC.

## 5. Conclusions

This study aimed to furnish a complete overview of QoL and SF aspects in CC patients. As previously reported, the available literature presents some discrepancies in terms of the heterogeneity of studies, the small number of cases, and the use of different questionnaires; thus, even the different study results are not always in accordance, making it impossible to reach definitive conclusions.

However, despite the limitations, it was possible to reach certain conclusions:

First of all, the evidence confirms the importance of a multidisciplinary approach for CC patients, which should focus, in a first phase, on understanding the patient’s characteristics and expectations, with the aim of tailoring the best treatment options for them. The second phase should be aimed at offering patients the best psychological support, with the goal of understanding their needs and preventing them from feeling alone in their fight against cancer. This phase should even be supported by a balanced diet and physical activities, which are an integral part of oncological and psychological therapies.

Psychological support remains crucially important for all oncologic patients. Several studies have demonstrated that psychological support has a positive impact on treatment adherence and oncologic outcomes [107,108].

Third, the therapeutic approach should be tailored not only based on pathology, but also on patient characteristics and expectations, because, as demonstrated in some studies [109,110], some strategies could have a more significant impact than others, and this should always be considered. Patients should always be involved in the decision process when choosing the therapeutic approach during counselling, and they should be guided through the process of considering the advantages and disadvantages of a particular therapy.

Fourth, it is necessary to conduct prospective studies with validated questionnaires in the near future in order to evaluate the impact and possible benefits on SF and QoL of different treatments. These studies should investigate all possible factors involved in potential changes of SF. Several investigations should be conducted on surgical techniques and approaches (such as laparoscopy, laparotomy, and robotic) and the impact of different treatment modalities (surgery, radiation therapy, and chemotherapy) on sexual function.

Future perspectives on CC treatment, including the improvement of QoL and SF, should be directed towards overcoming the pathology through cancer prevention, with the aim of eradicating a disease that has a significant oncological and social impact and often affects young patients.

## Figures and Tables

**Figure 1 medicina-59-00704-f001:**
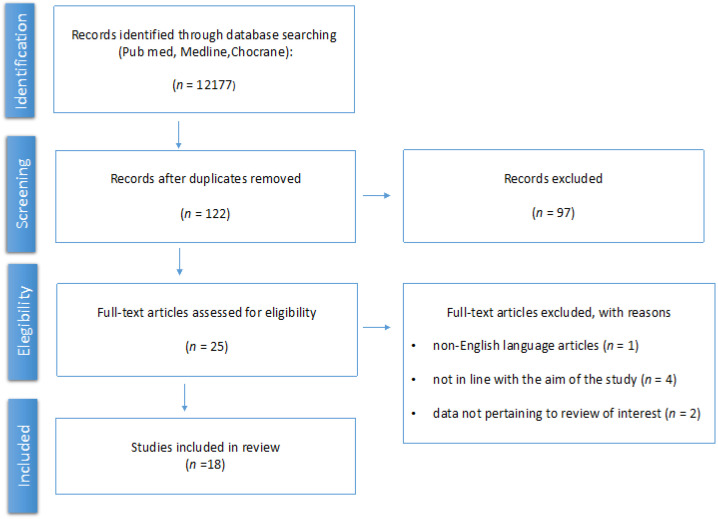
PRISMA flowchart for research strategy for the selected studies.

**Table 1 medicina-59-00704-t001:** The included studies and their characteristics.

Author	Study Design	Number of Patients and Features	Purpose	Questionnaire	Study Findings
Kirchheiner [38] 2021	Prospective study	1045 pts	SF in CC	EORTC QLQ-CX 24	60% were sexually active after treatment; Vaginal symptoms were negatively correlated with sexual enjoyment (*p* ≤ 0.001).
Novackova [39] 2022	Prospective study	36 pts	SF in early stage CC with nerve sparing radical hysterctomy therapy	FSFI EORTC QLQ-CX 24 EORTC QLQ-C30	Decreased sexual functioning after surgery, but no changes in sexual activity, enjoyment of worry
Park [40] 2007	Retrospective case-control study	860 pts 494 CG	SF in CC	EORTC QLQ-C30	CC pts had more sexual disfunction compared to CG. CC pts who received both surgery and adjuvant therapy than surgery alone reported significantly worse sexual or vaginal problems
Jensen [41] 2003	Multicenter prospective case-control study	173 pts 328 CG	SF after RH in CC	SVQ EORTC	CC had more vaginal symptoms, 91% were sexually active before and after surgery, but with a decrease of frequency
Ljuca [42] 2010	Retrospective-prospective study	35 pts	SF before and after chemoradiotherapy in CC	EORTC QLQ-CX 24	Improvement of vaginal function after chemoradiotherapy, but no difference in SF before and after therapy
Bakker [43] 2011	Cross-sectional study	252 pts	SF in CC	FSDS EORTC QLQ-CX24 HADS MMQ	38% CC had sexual distress; it was associated with vaginal symptoms and body imagine concerns
Aerts [44] 2014	Prospective case-control study	31 pts 93 CG 93 TLH	Compare SF in CC who only underwent surgery RH vs. healthy control group and womaen who underwent TLH for benign gynecological disease	SSFS SSPQ	The CC compared to the healthy group had a high risk of sexual disfunction; No difference in SF for CC compared to women with benign conditions
Correa [45] 2015	Retrospective case control study	37 pts 37 CG	SF in CC	FSFI	CC decreased SF by 64.9%, vaginal stenosis by 59.5%, not sexually active was at 80%, and who were sexually active had sexual dysfunction
Lee [46] 2016	Cross-sectional, case-control study	104 pts CG 104	compare SF between sexually active CC and healthy women.	EORTC QLQ-C30 EORTC QLQ-CX24 FSFI	No difference in SF
Carter [47] 2010	Prospective study	52 pts: 33 RT + 19 RH	SF and QOL in early-stage cervical cancer undergoing RT or RH	FSFI	FSFI mean scores were below 26.55—the clinical cut-off; No difference in SF between group with RH and RT
Hofsjö [48] 2017	Retrospective case-control study	34 pts 37 CG	SF in CC	a questionnaire designed to assess sexual function	Reduced satisfaction in sex, lubrification, elasticity and vaginal length; No differences in interest in sex between the pts and the CG in physical and psychological wellbeing and level of anxiety and depression
De Rosa [49] 2017	Prospective study	52 pts	Evaluates the effect on VHI, QoL, and SF of ospemifene in CC survivors.	EORTC QLQ-C30	Improvement of body image, sexual enjoyment and SF
Plotti [50] 2017	Retrospective study	90 pts	SF in long-term CC survivor affected by LACC and treated with type C2/type III RH	EORTC QLQ-CX24 EORTC QLQ-C30	Good level of sexual enjoyment with a slight worsening of sexual activity
Bae [51] 2015	Cross-sectional study	137 Pts	To examine the level of sexual function, depression and quality of life in CC pts	FSFI HADS FACT-G	CC pts had low sexual functionand about 45.4% of them experienced more than a moderate level of depression. Also, pts with lower sexual function had lower QoL and higher levels of depression
Serati [52] 2009	Prospective case-control study	38 pts (20 laparoscopic RH, 18 laparotomic RH) 35 CG	SF after RH vs. a control group of healthy women SF after laparoscopic RH vs. laparotomic RH	FSFI	RH worsens SF, without any significative difference between laparoscopy and laparotomy
Shi [53] 2020	RCT	91 pts RH: intervention group *n* = 46 control group *n* = 45	To assess the efficacy of a nurse-led positive psychology intervention on sexual function, depression and subjective well-being among postop pts with early CC	FSFI	Sexual function, depression and subjective well-being were significantly improved in the intervention group
Cerentini [54] 2019	Prospective case-control study	88 pts	use of vaginal dilators after brachytherapy in CC.	EORTC QLQ-C30	vaginal dilators did not increase the dimension of thevagina
Mihai Stanca [55] 2022	Retrospective observational study	430 pts	QoL and SF in CC	EORTC QLQ-CX 24 QLQ-C30	Decreased sexual function, activity and enjoyment

CG = control group—Pts = Patients. SF = sexual function—SVQ = Sexual function-Vaginal changes Questionnaire. RCT = randomized controlled study—FSFI = Female Sexual Function Index. RH = radical hysterectomy—FSDS = Female Sexual Distress Scale. RT = radical trachelectomy—LACC = Local advanced cervical cancer. Postop = postoperative—QoL = Quality of life. BIS = Body Images Scale—EORTC QLQ-C30 (European Organization and Treatment of Cancer QLQ-C30. SSFS= Short Sexual Functioning Scale—EORTC QLQ-CX24 = Quality of Life Questionnaire for Cervical Cancer. SSPQ = Specific Sexual Problems Questionnaire—MMQ = Maudsley Marital Questionnaire.HADS = Hospital Anxiety and Depression scale—FACT-G = Functional Assessment of Cancer Therapy–General.

**Table 2 medicina-59-00704-t002:** The main reported reasons for sexual disfunction.

No interest in sex [39,40,41,42,43,44,46,49]
Physical problem (depression, anxiety, surgical scar) [40,43,44,46,47,51,52,54]
No partner [39,41,43,44,45]
Vaginal dryness [38,39,40,41,48,49,50,52,54]
Dyspareunia and pain [39,40,41,42,43,47,49,53]

## Data Availability

Not applicable.

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
