# Peer review of "Post Treatment Sexual Function and Quality of Life of Patients Affected by Cervical Cancer: A Systematic Review"

_medicina, 2023, doi:10.3390/medicina59040704_

Round 1

Reviewer 1 Report (New Reviewer)

Excellent review on cervical cancer patients' quality of life, but 2 of the newest and largest studies (one of them on 430 patients) published by Stanca M. in Cancers in 2022 and performed also on EORTC questionnaires, are missing. 

Author Response

We really appreciated the reviewer 1’s comment which can improve the quality of the article.

Thanks for the advice. The suggested study is very interesting and pertinent to our review. We have proceeded to include it in our survey

Reviewer 2 Report (New Reviewer)

Dear editor:

I read the study with all interest. The manuscript medicina-2242187 by Cianci Stefano et al., attempts to detail “Post treatment sexual function and quality of life of patients affected by cervical cancer: A systematic review”

Here is some concerns:

There are some typo error, file needs to be edited by a native speaker.

Abstract:

Please clarify the duration time of studies included.

** Introduction:

·         Please cite the most updated statistics: PMID: 36545760, DOI: 10.1002/cnr2.1756

·         Since HPV vaccination is one the most strategy in prevention of cervical cancer please address this issue in introduction PMID: 32106837, PMCID: PMC7045378, DOI: 10.1186/s12889-020-8371-z,

In 2014, was published an article entitled: A systematic review of quality of life and sexual function of patients with cervical cancer after treatment, Please explain how new your study is to this study: PMID: 25033255, DOI: 10.1097/IGC.0000000000000207, PMCID: PMC9504584, PMID: 36143900.

** Material and methods:

What was the strategy search?

What was the question of study?

Please revise the method section based on the “PRISMA checklist systematic review”.

Please explain the data extraction with more details.

Figure 1 should be revised.

** Result:

It was written 16 studies included in the study. But in figure it was written 1.  Which is correct?

It seems search is not complete and comprehensive. For example I can see there is not some articles, please see:  PMCID: PMC9961044, PMID: 36834444

Result is so difficult to follow, please make some categories and bring the result under those titles or categories.

What is the Table 2???? Reported the main reasons for sexual disfunction. By which study?

Round 2

Reviewer 2 Report (New Reviewer)

Many thanks for the revision.

Please:

All abbriviations in abstract before use should be introduced by full words. 

EORTC QLQ-C30, FSFI, HADS and FSDS. 

Figure 1. should be revised and be drawn based on PRISMA chart. 

HRT???

Author Response

This manuscript is a resubmission of an earlier submission. The following is a list of the peer review reports and author responses from that submission.

Round 1

Reviewer 1 Report

1. The title should read as "Post treatment sexual function and quality of life of patients affected by cervical cancer: A systematic review"

2. The abstract stars with acronyms, that is not right e.g QoL, SF, CC

3. Authors keep writing sexual dysfunction after cervical cancer. This is not right. It should be written as "sexual dysfunction after cervical cancer treatment.

4. In the introduction, authors started with a reference citation before the sentence. This is not right.

5. This study only summarises the findings from previous studies without giving any perspective on what are the gaps and discuss hoe to fill those gaps.

6. There are no new insights in this work. A review article should show new insights and future projection of the topic.

7. The study does not show what is not known in the field. Only gives a summary of previous findings.

8. This study does not suggest any new solutions for the SF + QL in CC  patients post treatment. Explanations and comparisons on how do these patients suffer it is already known from the previous studies.

9. The overall suggestions from this study have already been suggested from a number of previous studies.

10. The study conclusion is based on the general findings of the previous studies instead of the findings form the present review article.

Reviewer 2 Report

This research paper is a valuable review concerning Quality of life to cervix carcinoma patients, but need some revisions

1-     The English language need serioys corrections starting from the abstract all over the manuscript

Ex: in the abstract correct; ‘’Evidence Syntesis’’

all studies, correct to All studies

disfunction, correct to dysfunction and so on all over the manuscript

the authors must revise the manuscript several times and use language check programe or let a native English speaker revise it

2-     Authors must mention the psychological disorders associated with postoperative sexual dysfunctions

3-     The table 2; It must be replaced with illustrative diagram or simple drawing network instead of one column table

4-     Abbreviations used in the abstract should be mentioned first; ex: CC, QoL

Round 2

Reviewer 1 Report

This article is still not suitable for publication in my opinion. Firstly, it is poorly written with unnecessary language in a scientific article (for example... "according to the most recent data"......"thanks to the strengthening of screening programs"...e.t.c . Secondly, it has a lot of wrong technical words for example authors have written "hpv" in small letters and in a short form for the first time without showing what is the long form? Thirdly, in the manuscript, cervical cancer sometimes is written as CC and sometimes is written in full as cervical cancer, throughout the manuscript. 

Reviewer 2 Report

the authors improved the manuscript and corrected relevant mistakes, and made a good effort to deserve publication